# Effect of Five Driver’s Behavior Characteristics on Car-Following Safety

**DOI:** 10.3390/ijerph20010076

**Published:** 2022-12-21

**Authors:** Junjie Zhang, Can Yang, Jun Zhang, Haojie Ji

**Affiliations:** 1Hefei Innovation Research Institute, Beihang University, Hefei 230012, China; 2School of Electronic and Information Engineering, Beihang University, Beijing 100191, China

**Keywords:** car-following safety, desired safety margin, driver’s behavior characteristics, sensitivity analysis

## Abstract

Driver’s behavior characteristics (DBCs) influence car-following safety. Therefore, this paper aimed to analyze the effect of different DBCs on the car-following safety based on the desired safety margin (DSM) car-following model, which includes five DBC parameters. Based on the Monte Carlo simulation method, the effect of DBCs on car-following safety is investigated under a given rear-end collision (RECs) condition. We find that larger subjective risk perception levels can reduce RECs, a smaller acceleration sensitivity (or a larger deceleration sensitivity) can improve car-following safety, and a faster reaction ability of the driver can avoid RECs in the car-following process. It implies that DBCs would cause a traffic wave in the car-following process. Therefore, a reasonable value of DBCs can enhance traffic flow stability, and a traffic control strategy can improve car-following safety by using the adjustment of DBCs.

## 1. Introduction

Varying driving behaviors influence traffic flow patterns and cause the occurrence of stopping shock waves. This response leads to traffic flow instability, which may be a longstanding cause of crashes [1,2,3]. In the United States, the statistical results show that 32,999 people died in car crashes according to National Highway Traffic Safety Administration in 2010. In addition, 58,523 people died in road traffic accidents in 2014 [4]. Rear-end collisions (RECs) are the most common accident type in the world [5]. For instance, RECs of 35.9% occurred in Washington State. Refs. [6,7] show that China’s highway REC accidents accounted for about 40% of all traffic accidents. Generally speaking, three major elements of a transportation system, including vehicles, road infrastructure, and human factors, determine roadway crashes [8,9,10]. However, human factors are the main impact factor for traffic flow stability and RECs. Therefore, it is necessary to carry out analysis and research on driver’s behavior characteristics (DBCs) of the car-following process, which is conducive to understanding the mechanism of shock wave evolution to avoid RECs.

DBCs, such as instantaneous acceleration/deceleration, driver’s reaction time, and risk perception level in particular, are especially vulnerable to psychological state. Ahmed et al. [11] presented a thorough review of the literature on the car-following models, which mainly study the driving behavior of vehicles on a roadway system by micro-simulation. Tarko et al. [12] presented an important finding that there is a strong correlation between the frequency of deceleration rates and the observed crash rates. Similarly, Kim et al. also obtained the relationship between micro scale driving behavior and REC propensity [13], who also found a high crash rate when drivers exhibit a high deceleration rate. An extended REC risk model is proposed considering perception–response time of drivers by a modified negative binomial regression [14]. Moreover, research also shows that driver risk perception of driving behaviors is very important for driving ability, which can enable drivers to avoid REC risks in complex traffic scenes [15,16,17]. Feng et al. [18] showed that aggressive driving behavior is the crucial factor for the personality of the drivers and violation of other vehicles. Tu et al. [19] found that the heterogeneity among drivers has a significant effect on road safety. Chung et al. [20] considered actual DBCs to develop the acceleration model and the sensitivity term for preventing RECs in the car-following process. This research shows that DBCs have important effects on the RECs. Moreover, micro-scale driver behaviors play important roles in the traffic flow pattern [21,22,23]. The traffic flow consists of the different drivers existing in the real traffic. Therefore, different drivers mean different DBCs in the car-following process. Basically, the occurrence of RECs caused by traffic shock waves in the context of heterogeneous traffic flow. Hourdos [24] pointed out that stopping shock waves causes RECs in the car-following process. Zheng et al. [25] also pointed out that an important risk factor of traffic accident is the variability in speed when successive shock waves happened on the freeway. Chatterjee and Davis [26] investigated a mechanism of REC on a crowded highway. Moreover, the stability theory of the car-following model shows that traffic shock waves spread backward, and leads to a higher REC risk for the vehicles behind the platoon [27]. Therefore, how to get a clear understanding of the DBCs in car-following safety becomes a topic worthy of much discussion.

Recently, some scholars have paid a lot of attention to the heterogeneous traffic flow [28,29,30,31,32,33]. However, this research is mainly concerned with the traffic flow stability analysis frameworks. Few studies have been focused on car-following safety analysis caused by different DBCs in heterogeneous traffic flow. Therefore, our aims are to analyze the influence of different DBCs on the car-following safety by using a microscopic car-following model. As discussed in previous literature, driver’s risk perception levels largely affected DBCs in the actual traffic. These DBCs mainly include physiological and psychological characteristics of drivers. The parameters of desired safety margin (DSM) model by Lu et al. [34] can characterize physiological and psychological characteristics of drivers, including driver’s reaction time, acceleration and deceleration preference coefficients, and upper and lower limits of the DSM. From this perspective, this study will employ the DSM model to investigate the influence of DBCs on the car-following safety.

To this end, other chapters of this paper are as follows. In Section 2, objectives and contributions of this study are given. In Section 3, the DSM model is introduced, and the corresponding DBCs parameters are calibrated. Section 4 analyzes the mechanism of REC in the car-following process, and REC conditions are derived. In Section 5, five DBCs parameters for REC risk are investigated according to the REC conditions. Numerical experiments and discussion are given in Section 6. In Section 7, our conclusions are given.

## 2. Objectives and Contributions

The objective of this study is to investigate the impact of DBC variables on the car-following safety. This study answers three questions regarding the car-following safety. (1) How do the response time, acceleration and deceleration preference coefficients, and the limits of the DSM affect the car-following safety? (2) How do different DBCs impact the REC risk? (3) How are the REC probability patterns related to the driver’s category in car-following process? The contributions of this study are two-fold. First, this study is a pioneering work exploring the quantitative relationship between the subjective risk perception levels, acceleration and deceleration sensitivity, reaction ability of the driver, and the car-following safety based on the Monte Carlo simulation method. Second, a reasonable value of DBCs can reduce traffic accidents and enhance traffic flow stability. Therefore, exploring the influence of some DBC variables on car-following safety is useful to rear-end risk.

## 3. DSM Model and Its Parameter Calibration

The DSM model by Lu et al. [35] is described as:
(1)$$a_{n}(t+\tau )=\left\{ \begin{array}{l} \alpha_{1}\left(S_{M}{}_{n}(t)-S_{M}{}_{nDH}\right),S_{M}{}_{n}(t)>S_{M}{}_{nDH}\\ \alpha_{2}\left(S_{M}{}_{n}(t)-S_{M}{}_{nDL}\right),S_{M}{}_{n}(t)<S_{M}{}_{nDL}\\ 0,else \end{array}\right.$$
and
(2)$$S_{M}{}_{n}(t)=1-\frac{V_{n}(t)\cdot \tau_{2}+{\left[V_{n}(t)\right]}^{2}/2d_{n}(t)}{X_{n-1}(t)-X_{n}(t)-L_{n-1}}+\frac{{\left[V_{n-1}(t)\right]}^{2}/2d_{n-1}(t)}{X_{n-1}(t)-X_{n}(t)-L_{n-1}}$$
where *a_n_*(*t*) is the *n*th vehicle’s acceleration; *S_MnDH_* and *S_MnDL_* are respectively the lower and upper limits of the DSM for the *n*th driver; *α*_1_ and *α*_2_ denote the sensitivity coefficients of acceleration and deceleration, respectively; *τ* is driver’s reaction time; *τ*_2_ is the brake system’s reaction time, and its value sets 0.15 s for the car; *V_n_*(*t*) is the *n*th vehicle’s speed; $\Delta X_{n}(t)=X_{n-1}(t)-X_{n}(t)$ denotes spacing headway; *d_n_*(*t*) is the *n*th vehicle’s deceleration; *L*_*n*−1_ is the length of the *n*−1th vehicle.

DBCs parameters based on DSM model have been calibrated to use Genetic Algorithm using enhanced NGSIM dataset. We choose sixty car-following cases. Table 1 shows that the statistic results of DBCs parameters. From Table 1, we obtain that the mean of *α*_1_ is about 8.51 m/s^2^, the std. deviation of *α*_1_ is about 5.301 m/s^2^, and that the mean of *α*_2_ is about 14.14 m/s^2^ and the std. deviation of *α*_2_ is about 8.276 m/s^2^, and that the mean of *S_MnDL_* is 0.76, and the std. deviation of *S_MnDL_* is 0.125, and that the mean of *S_MnDH_* is 0.95, and the std. deviation of *S_MnDH_* is 0.045. The mean of driver’s reaction time *τ* is 0.65 s, and the std. deviation of *τ* is 0.297 s.

Figure 1 shows that different DBCs are obtained by using K-nearest neighbor clustering analysis of five parameters. Among them, acceleration and deceleration preference coefficients are divided into three kinds: sensitive, moderate sensitivity and insensitive; upper and lower limits of the DSM are divided into risk-averse, risk-neutral, and risk-prone. In addition, driver’s reaction time is divided into responsive, moderate response and unresponsive. Moreover, most DBCs are responsive, moderately sensitive, and risk-prone from Table 2, which is in accordance with DBCs under actual car-following status.

Therefore, investigating the influence of DBCs on the RECs risk by using the DSM car-following model is a topic worthy of much discussion.

## 4. Results for the REC Mechanism Car-Following Process

In the braking process, the *n*th and *n*−1th vehicles initially move at a *v*_0_ and initiate braking at decelerations of *a*_*n*−1_ and *a_n_*, respectively, as shown in Figure 2.

Then, a REC is avoided when the sum of car-following safety distance needed by the leading vehicle is more than the following vehicle’s stopping distance as follows:
(3)$$v_{0}\cdot \tau +v_{0}^{2}/2a_{n}^{}\leq v_{0}^{2}/2a_{n-1}^{}+\Delta S_{n}^{n-1}$$
where $\Delta S_{n}^{n-1}$ is car-following safety distance; *τ* is following the driver’s reaction time.

The deceleration of following vehicle should be satisfied to avoid rear-end collision as


(4)
$$a_{n}\geq \frac{v_{0}^{2}}{v_{0}^{2}/2a_{n-1}^{}+2\left(\Delta S_{n}^{n-1}-v_{0}\tau \right)}$$


Then, the stopping distance available to the following vehicle is given


(5)
$$S_{n}=v_{0}^{2}/2a_{n}^{}+\Delta S_{n}^{n-1}-v_{0}\tau$$


We note that the driver adjusts acceleration/deceleration according to homeostatic risk perception during their actual driving. Therefore, time headway (TH) can be derived as follows:


(6)
$$TH_{t}=\frac{\tau_{2}}{1-\chi }+\frac{L_{n-1}}{v_{0}},$$


Then, car-following safety distance $\Delta S_{n}^{n-1}$ is derived based on Equation (6) as
(7)$$\Delta S_{n}^{n-1}=\frac{\tau_{2}v_{0}}{1-\chi }+L_{n-1},$$
where $\chi =\langle S_{M}{}_{nDL},S_{M}{}_{nDH}\rangle$, 〈⋅〉 denotes an accepted value to ensure car-following safety.

Equations (4) and (6) indicate that the following car needs a bigger deceleration to avoid collision when the driver’s reaction time *τ* is longer than his/her *TH_t_*. It implies that driver’s sensitivity factor for acceleration/deceleration of the leading car affects deceleration of the following car. Furthermore, Equation (7) implies that *S_MnDL_* and *S_MnDH_* determine time headway in the car-following process. In the platoon, the deceleration of leading vehicle denotes *a*_1_ considering individual DBCs when a stopping wave is introduced.

Then, the available stopping distance of car 2 is obtained by


(8)
$$S_{2}=v_{1}^{2}/2a_{1}+\frac{\tau_{2}v_{2}}{1-\chi_{2}}+L_{1}-v_{2}\tau_{2}$$


Similarly, the available stopping distance of the *N*th car in the platoon is obtained by:
(9)$$S_{N}=v_{1}^{2}/2a_{1}+\sum_{i=2}^{N}\left(\frac{\tau_{2}v_{i}}{1-\chi_{i}}+L_{i-1}-v_{2}\tau_{i}\right)-\left(N-2\right)S_{0}$$
where *S*_0_ denotes the minimum safe stopping distance among two cars. The relative stopping distance between two cars varied from 1.07 m to 4.83 m, and its mean is 2.17 m [31].

According to Equation (9), we obtain REC condition of the *N*th car as:
(10)$$\sum_{i=2}^{N}\left(\frac{\tau_{2}v_{i}}{1-\chi_{i}}+L_{i-1}-v_{2}\tau_{i}\right)>v_{1}^{2}/2a_{1}-v_{N}^{2}/2a_{\min}-\left(N-2\right)S_{0}$$
where *a*_min_ is a maximum deceleration. As suggested by the Highway Capacity Manual, the deceleration of a car varied from 2 and 8 m/s^2^ [36]. Thus, *a*_min_ is set to be −8 m/s^2^ in this study.

## 5. Sensitivity Analysis of Five DBCs on REC Risk

According to the previous discussion, the REC condition is related to DBC parameters. Therefore, we conduct a vehicle platoon that 20 vehicles move on a straight road with initial space headway *S_H_* = 40 to numerically illustrate the influence of five DBC parameters on REC risk, as shown in Figure 3. In addition, the initial positions and speeds of 20 vehicles are showed as follows:


(11)
$$x_{n}(0)=S_{H}\cdot N,v_{n}(0)=20,{\dot{v}}_{n}(0)=0,n=2,\cdots ,N.$$


In the platoon, the leading car brakes at –1.5 m/s^2^ after 200 s until it is stopped. In addition, other following cars follow the leading car to decelerate until they are stopped. Other parameters of model are set as as *τ*_2_ = 0.15 s, *d_n_*(*t*) = *d*_*n*−1_(*t*) = 7.35 m/s^2^, *L_n_* = 4.3 m, *n* = 1, 2, 3, …, *N*. Collision happening is to determine whether two successive vehicles collide or not at any moment in the car-following process according to REC conditions of Equation (10). 

The numerical experiment is presented using the Monte Carlo simulation method to estimate the probability of REC in the traffic condition described above. Based on the statistical results of 10,000 samples, we investigate the probability of REC $P(C_{n-1}^{n})$ under DBC parameters effect.

### 5.1. Impact of Response Time

According to the calibration results of Table 1, the range of response time is set as τ∈[0.3,1.6] s. Other driving behavior parameters with mean and std. deviation are set as: the acceleration and deceleration preference coefficients $\alpha_{1}=8.51\pm 5.301$ m/s^2^ and $\alpha_{2}=14.14\pm 8.276$ m/s^2^, and the limits of the DSM $S_{M}{}_{nDL}^{}=0.73\pm 0.132$ and $S_{M}{}_{nDH}^{}=0.94\pm 0.056$. Ten drivers of the platoon are randomly arranged on a single lane. 

The varying of REC probability with response time is shown in Figure 4. From Figure 4, the REC risk is increasing with *τ*. It implies that the stopping waves gradually increase as response time goes on to cause the increasing of the REC probability of vehicles. Car-following safety can improve with the decrease in the driver’s response time. Thus, drivers need to keep faster reaction ability to avoid the occurrence of REC.

### 5.2. Impact of the Limits of the DSM

As shown in previous analysis, acceleration and deceleration preference coefficients are set as $\alpha_{1}=8.51\pm 5.301$ m/s^2^ and $\alpha_{2}=14.14\pm 8.276$ m/s^2^, respectively. Driver’s reaction time is set as τ=0.65±0.297 s. In order to investigate the influence of *S_MnDH_* on the REC probability, the range of *S_MnDH_* is set as $S_{M}{}_{nDH}^{}\in \left[0.86,1\right]$ to ensure $S_{M}{}_{nDH}^{}\geq S_{M}{}_{nDL}^{}$, in which $S_{M}{}_{nDL}^{}=0.73\pm 0.13$. Moreover, the range of *S_MnDL_* is set as $S_{M}{}_{nDL}^{}\in \left[0.73,0.88\right]$ to ensure $S_{M}{}_{nDL}^{}\leq S_{M}{}_{nDH}^{}$, in which $S_{M}{}_{nDH}^{}=0.94\pm 0.056$.

The varying of REC probability with the limits of the DSM is shown in Figure 5. From Figure 5a, the REC probability of vehicles decreases with the increasing of *S_MnDH_*. However, the REC probability of vehicles has no obvious reduction when *S_MnDH_* is increased to a certain value (*S_MnDH_* ≥ 0.96) in our case study. Likewise, the REC probability of vehicles decreases with the increasing of *S_MnDL_* as shown in Figure 5b. However, the REC risk has more sensitivity to *S_MnDL_*. The REC accidents will not happen when *S_MnDL_* ≥ 0.85 in our case study. Results imply that the stopping waves gradually weaken with the increasing of the *S_MnDL_* or *S_MnDH_*. Thus, a large upper (or lower) limit of the DSM can reduce the REC risk. Adjusting the limits of the DSM of the driver can improve traffic safety in the car-following process.

### 5.3. Impact of Acceleration and Deceleration Preference Coefficients

The limits of the DSM are set as $S_{M}{}_{nDL}^{}=0.73\pm 0.132$ and $S_{M}{}_{nDH}^{}=0.94\pm 0.056$, respectively. In addition, the response time is τ=0.65±0.297 s. To analyze the impact of the acceleration and deceleration preference coefficients on the REC probability, the range of acceleration preference coefficient is set as $\alpha_{1}\in \left[3,30\right]$ m/s^2^, in which the deceleration preference coefficient $\alpha_{2}=14.14\pm 8.276$ m/s^2^. However, the range of deceleration preference coefficient is set as $\alpha_{2}\in \left[9,30\right]$ m/s^2^, in which the acceleration preference coefficient $\alpha_{1}=8.51\pm 5.301$ m/s^2^.

The varying of REC probability with the acceleration and deceleration preference coefficients is shown in Figure 6. The REC probability of vehicles increases with the increasing of the acceleration preference coefficient as shown in Figure 6a. However, the REC probability of vehicles decreases with the increasing of the deceleration preference coefficient from Figure 6b. In addition, the REC probability of vehicles has no obvious reduction when the deceleration preference coefficient is increased to a certain value (*α*_2_ > 18 m/s^2^) in our case study. Results show that a large deceleration preference (or a small acceleration preference) can reduce the REC risk, and that the stopping waves gradually weaken with increasing of the deceleration preference and decreasing of the acceleration preference. Therefore, the car-following safety can improve if the driver can maintain the smaller acceleration preference and the larger deceleration preference for drivers.

As described previously, findings further verify that five DBCs parameters play an important role in the traffic waves, thereby causing REC risk in the car-following process. The choice of interval DSMs, the acceleration and deceleration preference coefficients, and the driver’s response time influence the REC probability. It implies that a reasonable choice of fiver DBCs parameters can be provided to reduce the REC risk.

## 6. Numerical Experiment and Discussion

In our numerical simulation, the initial conditions of speeds and positions for all vehicles in the platoon are as follows:
(12)$$\left\{ \begin{array}{l} X_{1}(0)=L\cdot N,V_{1}(0)=20,{\dot{V}}_{1}(0)=0,\\ {\dot{V}}_{1}(t)=\xi_{1}(\tilde{t}),t\geq \tilde{t},\\ X_{n}(0)=L\cdot (N-n+1),V_{n-1}(0)=20,{\dot{V}}_{n-1}(0)=0,n=2,\cdots ,N-1, \end{array}\right.$$
where spacing headway *L* is 35 m; the number of vehicles in the platoon *N* is 50; *X_n_*(0) denotes the *n*th vehicle’s initial position; *V_n_*(0) is the *n*th vehicle’s initial speed; ${\dot{V}}_{n}(0)$ is the *n*th vehicle’s initial acceleration; and $\xi_{1}(\tilde{t})$ denotes a small acceleration disturbances of the leading car after $\tilde{t}$, and its distribution function obeys $5\times 10^{-2}\times U(-1,1)$.

According to Table 2, DBCs have 3^3^ kinds of combinations based on driving behavior parameters. The vector $\Omega_{i}=\left(\tau ,\alpha_{1},\alpha_{2},SM_{nDH},SM_{nDL}\right),i=1,2,\cdots M$ is used to represent DBCs. In this study, in order to analyze the impact of heterogeneity DBCs on REC risk, we choose four kinds of combinations of DBCs parameters without the loss of generality as follows:


$$\begin{array}{l} \Omega_{1}=\left(0.4,7.27,13.59,0.91,0.89\right),\\ \Omega_{2}=\left(0.4,12.56,23.09,0.95,0.9\right),\\ \Omega_{3}=\left(1.1,7.27,13.59,0.84,0.56\right),\\ \Omega_{4}=\left(0.7,6.82,6.64,0.94,0.75\right), \end{array}$$


We will exhibit four cases of the different combinations for four kinds of DBCs. Four kinds of DBCs in the platoon are arbitrary. Four cases are conducted as below:Case 1: ℂ_1_ of 50% and ℂ_2_ of 50%;Case 2: ℂ_3_ of 50% and ℂ_4_ of 50%;Case 3: ℂ_1_ of 90%, ℂ_2_ of 4%, ℂ_3_ of 2% and ℂ_4_ of 4%;Case 4: ℂ_1_ of 70%, ℂ_2_ of 26%, ℂ_3_ of 2% and ℂ_4_ of 2%;Case 5: ℂ_1_ of 30%, ℂ_2_ of 50%, ℂ_3_ of 10% and ℂ_4_ of 10%;Case 6: ℂ_1_ of 10%, ℂ_2_ of 20%, ℂ_3_ of 40% and ℂ_4_ of 30%;Case 7: ℂ_1_ of 10%, ℂ_2_ of 10%, ℂ_3_ of 46% and ℂ_4_ of 34%;Case 8: ℂ_1_ of 6%, ℂ_2_ of 6%, ℂ_3_ of 50% and ℂ_4_ of 38%;


Figure 7 depicts gap distributions simulated by a DSM model with four cases of different DBCs obtained at *t* = 300, *t* = 500 s, *t* = 800 s, and *t* = 1000 s. The amplitude fluctuation of gap becomes large with the increase in Ω_3_ and Ω_4_. Here, Ω_3_ and Ω_4_ can be claimed as unstable drivers, and Ω_1_ and Ω_2_ can be claimed as stable drivers. Therefore, it implies that the shock wave occurs, and the RECs happen when Ω_3_ and Ω_4_ increase in the platoon as shown in the cases 3–4 of Figure 7a–d. Otherwise, decreasing the proportions of Ω_3_ and Ω_4_, the shock waves gradually weakened, and REC will not occur as shown in cases 1–2 of Figure 7a–d.

Furthermore, the safety margin (SM) is introduced as a risk indicator to analyze car-following safety, its mathematical expression is as follows:


(13)
$$S_{M}{}_{n}=1-\frac{0.15\cdot V_{n}(t)+{\left[V_{n}(t)\right]}^{2}/1.5g}{\Delta X_{n}(t)-L_{n-1}}+\frac{{\left[V_{n-1}(t)\right]}^{2}/1.5g}{\Delta X_{n}(t)-L_{n-1}}$$


Moreover, a probabilistic measure is introduced to estimate REC risk in the car-following process as:
(14)$$P^{\left(t\right)}(C_{n-1}^{n})={\mathrm{Pr}}^{\left(t\right)}(C_{n-1}^{n}|S_{M}{}_{n-1}^{n})=\exp\left(-\frac{1}{c}\cdot S_{M}{}_{n-1}^{n}\right)$$
where $P^{\left(t\right)}(C_{n-1}^{n}|S_{M}{}_{n-1}^{n})$ is the probability that the *n* vehicle collides with the *n*−1 vehicle; *c* denotes constant argument, and its value depends on road characteristics. Here, *c* is set as 0.25 in our numerical simulation.

Figure 8 shows REC probability patterns of all other cars except for the leading car based on the SM under different DBCs. RECs are a small probability event as shown in Figure 8a,b. However, rear-end collision probability of car-following will increase gradually compared with Figure 8a,b and Figure 8c,d. Moreover, unstable DBCs can enhance the traffic wave and increase REC risk as shown in Figure 8e–h. Otherwise, stable DBCs can attenuate the shock wave and reduce REC risk. Results show that different DBCs would lead to shock waves, thereby increasing REC probability in the car-following process.

To summarize, results show that a potential strategy of the adjustment of the proportions of the DBCs can attenuate the shock wave and avoid RECs, and that the decreasing of the proportions of stable DBCs causes the RECs.

## 7. Conclusions

DBCs play an important role in car-following safety. Different DBCs may lead to unstable traffic flow, and forming a shock wave would cause RECs under certain traffic conditions. Therefore, exploring DBCs in car-following safety is a valuable research issue. The DBCs are portrayed by using microscopic car-following models. In our study, we used the DSM car-following model to analyze the influence of DBCs on the car-following safety. The DSM model uses *S_MnDL_* and *S_MnDH_* to describe driving risk preference; *α*_1_ and *α*_2_ to describe the acceleration and deceleration sensitivity; and *τ* to describe driver’s responsiveness. Therefore, the risk preference, sensitivity, and responsiveness affect car-following safety.

To quantify the DBCs, five parameters are calibrated by using sixty cases from NGSIM datasets. We obtain REC conditions by using the DSM model, and analyze the probability of REC under those parameters’ effect by using the Monte Carlo method. Findings show that the decreasing of response time can improve traffic safety, and that a large *S_MnDL_* or *S_MnDH_* can reduce the REC risk, and that the traffic safety can improve if the driver can maintain the smaller acceleration sensitivity and the larger deceleration sensitivity. Because these parameters can portray DBCs, each parameter divided into three types of DBCs to further discuss the impact of different DBCs on the REC risk. Through the numerical experiments, if the vehicle platoon has stable and unstable DBCs, then the proportion of drivers with different DBCs plays an important role in the car-following safety. Once the traffic flow is in an unstable state, the shock waves gradually enhanced, and REC will occur. Moreover, REC probability patterns also imply that different DBCs with an inappropriate proportion would lead to shock waves, thereby increasing REC risk. A potential strategy of the adjustment of the proportions of the unstable DBCs can improve car-following safety.

In summary, the DBCs affect the car-following safety. The findings are useful in setting a reasonable proportion of the unstable and stable DBCs to stabilize traffic flow, and developing traffic control strategy for REC prevention by using the adjustment of DBCs.

## Figures and Tables

**Figure 1 ijerph-20-00076-f001:**
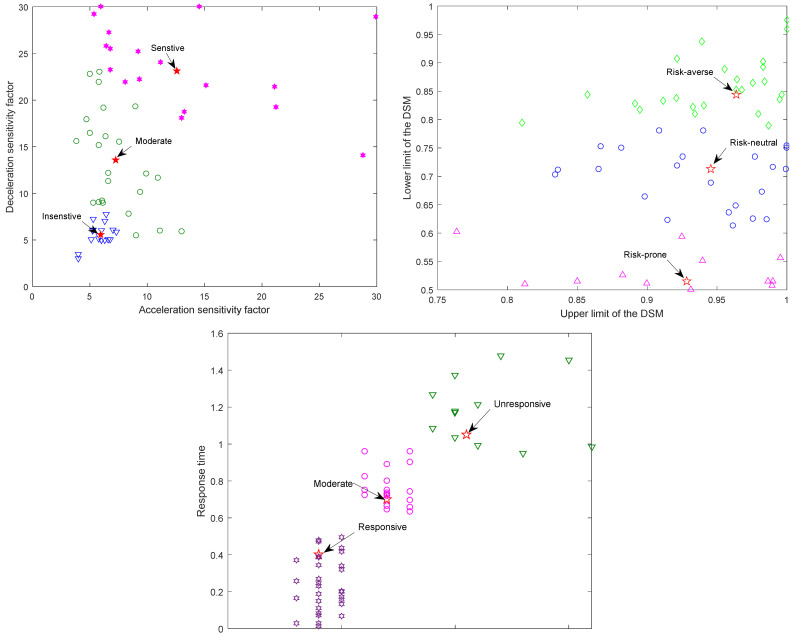
Classification of DBCs.

**Figure 2 ijerph-20-00076-f002:**
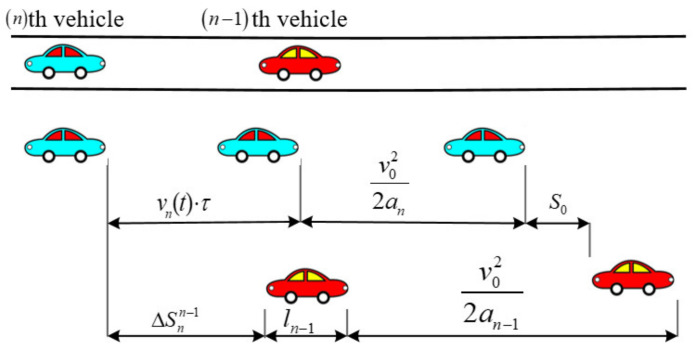
Braking process between two successive vehicles during a car-following situation.

**Figure 3 ijerph-20-00076-f003:**
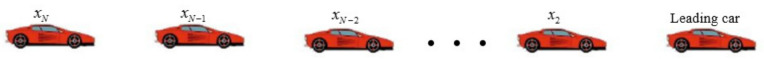
Cars moving on a straight road.

**Figure 4 ijerph-20-00076-f004:**
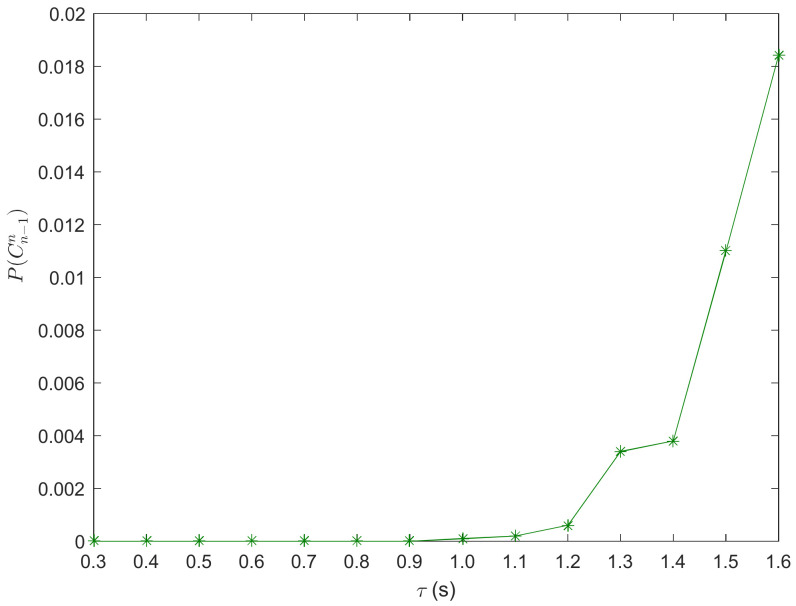
Varying of REC probability with response time.

**Figure 5 ijerph-20-00076-f005:**
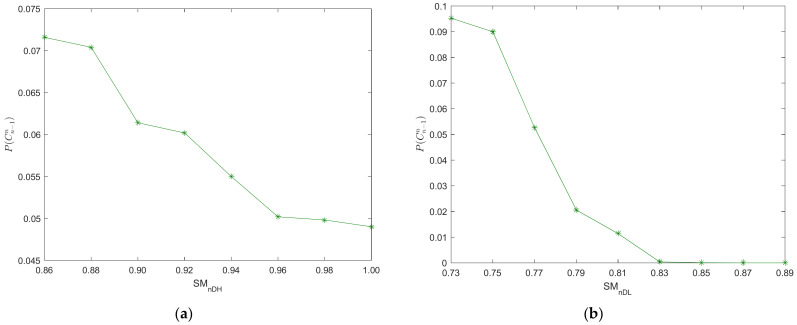
Varying of REC probability with the limits of the DSM, (**a**) upper limit of the DSM; (**b**) lower limit of the DSM.

**Figure 6 ijerph-20-00076-f006:**
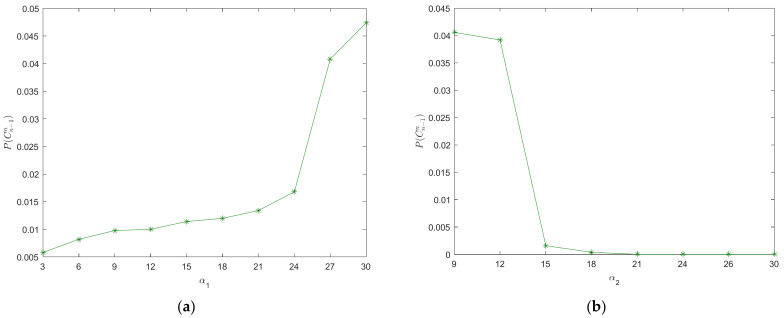
Varying of REC probability with the acceleration and deceleration preference coefficients, (**a**) acceleration preference coefficient; (**b**) deceleration preference coefficient.

**Figure 7 ijerph-20-00076-f007:**
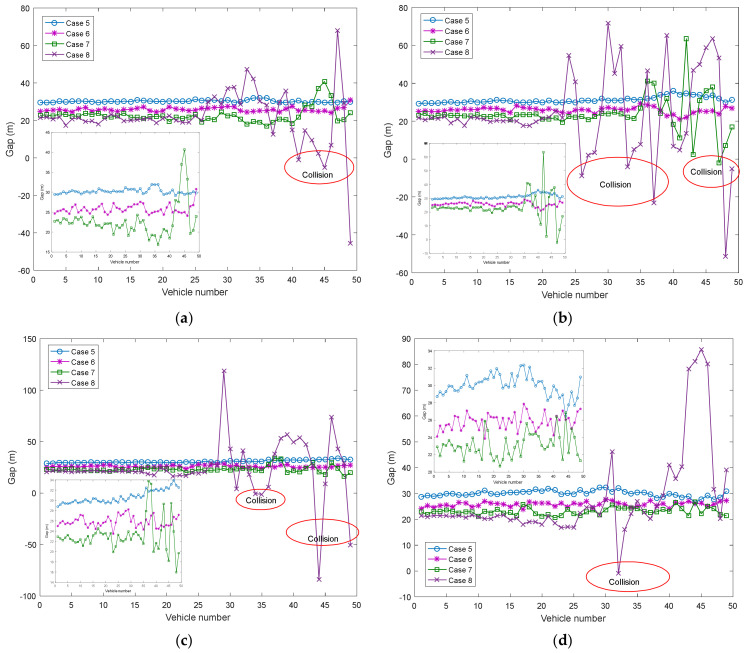
Gap patterns of all vehicles with DBCs at: (**a**) *t* = 300 s; (**b**) *t* = 500 s; (**c**) *t* = 800 s; (**d**) *t* = 1000 s.

**Figure 8 ijerph-20-00076-f008:**
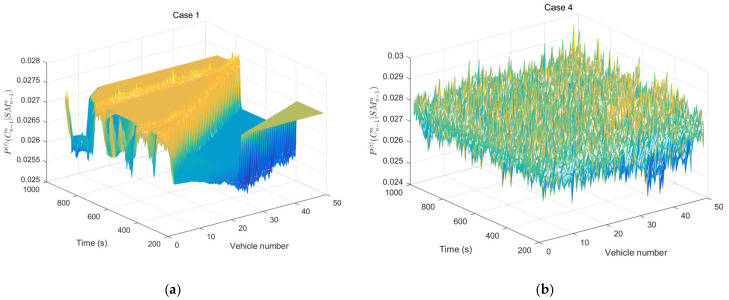
REC probability patterns of all other cars (except the leading car) under different DBCs.

**Table 1 ijerph-20-00076-t001:** Statistic results of DBCs.

Parameters	Mean	Std. Deviation	Median	Minimum	Maximum
*α* _1_	8.51	5.301	6.52	3.79	29.91
*α* _2_	14.14	8.276	12.18	3.01	30.00
*S_MnDL_*	0.73	0.132	0.75	0.50	0.98
*S_MnDH_*	0.94	0.056	0.94	0.76	1
*τ*	0.65	0.297	0.48	0.30	1.60

**Table 2 ijerph-20-00076-t002:** Clustering results of DBCs.

Parameters	DBCs	Observation	Mean	Std. Deviation	Median
*α* _1_	Insensitive	18	5.92	0.909	5.60
*α* _2_			5.57	1.173	5.28
*α* _1_	Moderate	24	7.27	2.350	6.27
*α* _2_			13.59	5.418	12.18
*α* _1_	Sensitive	18	12.56	7.668	10.24
*α* _2_			23.09	4.489	23.64
*S_MnDL_*	Risk-averse	12	0.53	0.035	0.52
*S_MnDH_*			0.91	0.075	0.93
*S_MnDL_*	Risk-neutral	23	0.70	0.053	0.71
*S_MnDH_*			0.94	0.053	0.95
*S_MnDL_*	Risk-prone	25	0.86	0.049	0.84
*S_MnDH_*			0.95	0.048	0.96
*τ*	Responsive	30	0.42	0.068	0.4
	Moderate	18	0.71	0.076	0.7
	Unresponsive	12	1.13	0.227	1.1

## Data Availability

Driving behavior data used to support the findings of this research are available from the first author upon request.

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
