# Peer review of "Effect of Five Driver’s Behavior Characteristics on Car-Following Safety"

_ijerph, 2022, doi:10.3390/ijerph20010076_

Round 1

Reviewer 1 Report

This paper investigated the impact of five DBCs in car-following behavior on the rear-end risk. For this purpose, the authors assumed that a DSM model would represents the car-following process well. The correlation coefficients of this model characterize driver characteristics to analyze REC probability in car folllowing process. Overall, this paper is a good contribution to the existing literature. Also please make few minor revisions according to my comments below.

1. The literature review is a bit outdated. More recent research should be recognized including but not limited to

Wang, L., Wu, J., Li, X., Wu, Z. and Zhu, L. (2022) ‘Longitudinal control for person-following robots’, Journal of Intelligent and Connected Vehicles, 5(2), pp. 88–98. doi:10.1108/jicv-01-2022-0003.

Ahmed, H. U., Huang, Y., & Lu, P. (2021). A review of car-following models and modeling tools for human and autonomous-ready driving behaviors in micro-simulation. Smart Cities, 4(1), 314-335.

2. My overall concern with the paper was understanding clearly how you would calibrate this model and develop your alpha_1, alpha_2, and tau. 

3. Why was tau-2 not varied in Equation 2? The brake response times of trucks and cars are different.

4. Why not calibrate the model separately for cars and trucks in the NGSIM data? What are the differences in rear-end crash risk based between cars and trucks based on your calibration method? Please briefly explain in this case. 

Author Response

Dear  Reviewer,

First of all, we would like to express our gratitude to the editors for giving us the opportunity to revise our manuscript entitled “Effect of five driver’s behavior characteristics on car-following safety” (ijerph-2099462). We also deeply appreciate the reviewers’ insightful and constructive comments. These comments are extremely helpful in revising and improving our paper. According to the reviewers’ comments, we have significantly revised the original manuscript. Please refer to the attachment for specific modification comments.

We believe that the revised manuscript has been significantly improved. We would appreciate it if the reviewers can take another look at the revised manuscript and provide more valuable comments/suggestions.

Once again, we appreciate your time and consideration.

Sincerely,

Junjie Zhang

Reviewer 2 Report

The authors investigated the effect of five DBCs on car-following safety and conducted simulation experiments to evaluate the sensitivity of five DBCs on REC risk. Overall, the paper is well-written and makes significant contributions from the practical implication viewpoints. Before the publication recommendation, I would appreciate it if the authors could address the following minor revisions.

1. The authors are suggested to make a clear statement on the contribution of this study.

2. Please briefly explain theSMrisk indicator, why you used it (section 5)? and why not use the TTC and TH as risk indicator?

3. Please briefly illustrate the clustering method (table 2).

4. Correcting minor grammatical errors/typos can improve readability.

Author Response

Dear Editors & Reviewers,

First of all, we would like to express our gratitude to the editors for giving us the opportunity to revise our manuscript entitled “Effect of five driver’s behavior characteristics on car-following safety” (ijerph-2099462). We also deeply appreciate the reviewers’ insightful and constructive comments. These comments are extremely helpful in revising and improving our paper. According to the reviewers’ comments, we have significantly revised the original manuscript. Please refer to the attachment for specific modification comments.

We believe that the revised manuscript has been significantly improved. We would appreciate it if the reviewers can take another look at the revised manuscript and provide more valuable comments/suggestions.

Once again, we appreciate your time and consideration.

Sincerely,

Junjie Zhang
